# Curcumin Inhibits TORC1 and Prolongs the Lifespan of Cells with Mitochondrial Dysfunction

**DOI:** 10.3390/cells13171470

**Published:** 2024-09-01

**Authors:** Arshia Naaz, Yizhong Zhang, Nashrul Afiq Faidzinn, Sonia Yogasundaram, Rajkumar Dorajoo, Mohammad Alfatah

**Affiliations:** 1Genome Institute of Singapore (GIS), Agency for Science, Technology and Research (A*STAR), Singapore 138672, Singapore; 2Bioinformatics Institute (BII), Agency for Science, Technology and Research (A*STAR), Singapore 138671, Singapore; 3Department of Paediatrics, Yong Loo Lin School of Medicine, National University of Singapore, Singapore 119228, Singapore; 4Healthy Longevity Translational Research Program, Yong Loo Lin School of Medicine, National University of Singapore, Singapore 117544, Singapore; 5Centre for Healthy Longevity, National University Health System, Singapore 117456, Singapore

**Keywords:** curcumin, postmitotic cells, aging, lifespan, mitochondria, TORC1, yeast

## Abstract

Aging is an inevitable biological process that contributes to the onset of age-related diseases, often as a result of mitochondrial dysfunction. Understanding the mechanisms behind aging is crucial for developing therapeutic interventions. This study investigates the effects of curcumin on postmitotic cellular lifespan (PoMiCL) during chronological aging in yeast, a widely used model for human postmitotic cellular aging. Our findings reveal that curcumin significantly prolongs the PoMiCL of wildtype yeast cells, with the most pronounced effects observed at lower concentrations, indicating a hormetic response. Importantly, curcumin also extends the lifespan of postmitotic cells with mitochondrial deficiencies, although the hormetic effect is absent in these defective cells. Mechanistically, curcumin inhibits TORC1 activity, enhances ATP levels, and induces oxidative stress. These results suggest that curcumin has the potential to modulate aging and offer therapeutic insights into age-related diseases, highlighting the importance of context in its effects.

## 1. Introduction

Aging is an inevitable biological process that affects all living organisms, marked by the gradual decline in cellular function and a heightened susceptibility to age-related diseases, such as cardiovascular diseases (CVDs), neurodegenerative diseases (NDDs), sarcopenia, and age-related macular degeneration (AMD) [1,2,3]. These diseases pose significant challenges not only to individual well-being but also to global healthcare systems, especially as the world’s population continues to age rapidly [4]. As a result, understanding the underlying mechanisms of aging and identifying potential interventions to delay or mitigate its effects have become key goals in biomedical research.

Mitochondrial dysfunction is recognized as a central contributor to the development of many age-related diseases [5,6,7,8]. Mitochondria, often referred to as the “powerhouses of the cell”, are essential for energy production, cellular metabolism, and the regulation of oxidative stress [9]. However, as organisms age, mitochondrial function deteriorates, triggering a cascade of detrimental events that contribute to disease pathogenesis [5,6,7,8,9]. In particular, mitochondrial dysfunction in postmitotic cells has been linked to the aging process and its associated diseases. For instance, in CVDs, impaired mitochondrial energy production and increased oxidative stress play critical roles in the development of cardiac pathologies [10,11,12]. Similarly, mitochondrial dysfunction is a hallmark of NDDs, including Alzheimer’s and Parkinson’s diseases, where it exacerbates neuronal damage and cell death [13,14]. Additionally, mitochondrial dysfunction has been implicated in sarcopenia, characterized by the age-related loss of muscle mass and function [15,16], as well as in AMD, a leading cause of vision loss in the elderly [17,18,19].

In recent years, natural compounds have gained attention as potential anti-aging interventions [20,21]. These compounds, derived from various plant sources, have shown promise in modulating biological pathways associated with aging, such as oxidative stress, inflammation, and mitochondrial function. As a result, they are being actively explored as therapeutic agents that can slow down the aging process, enhance lifespan, and improve healthspan—the period of life spent in good health.

Among these natural compounds, curcumin stands out as a particularly potent candidate for anti-aging interventions. Curcumin is a polyphenolic compound extracted from the rhizome of Curcuma longa (turmeric), a spice that has been used for centuries in traditional medicine. Extensive research has demonstrated curcumin’s wide-ranging therapeutic potential, including its ability to mitigate the effects of aging and increase lifespan [22,23,24,25]. However, despite its promising effects in experimental settings, there remains a critical gap in understanding the impact of dietary curcumin on cellular longevity.

The current study seeks to fill this gap by investigating the molecular and cellular mechanisms through which curcumin influences postmitotic cellular lifespan (PoMiCL) during chronological aging, with a specific focus on the role of mitochondrial dysfunction. Yeast, a well-established model organism for biomedical research including cellular aging [26,27,28,29,30,31], was employed to explore how curcumin affects PoMiCL, especially under conditions of compromised mitochondrial function. Our mechanistic findings shed light on the complex interplay between curcumin treatment, mitochondrial function, and PoMiCL, offering valuable insights into the potential utility of curcumin as an anti-aging modulator.

## 2. Results

### 2.1. Curcumin Extends Lifespan of Postmitotic Cells during Chronological Aging

We investigated the effect of curcumin treatment on the lifespan of yeast wildtype non-dividing cells within the context of chronological aging. We performed postmitotic cellular lifespan (PoMiCL) assays that provided us with a dynamic platform to meticulously dissect the impact of curcumin on cellular survival during the gradual progression of chronological aging. To initiate the experimental regimen, yeast wildtype cells were supplemented with various concentrations of curcumin, ranging from 12.5 µM to 200 µM, in a 96-well microplate. To initiate the chronological aging process, we first allowed the cells to transition into the postmitotic state. This was achieved by cultivating the cell cultures in medium until they reached the stationary phase, marking the beginning of the PoMiCL assays during chronological aging [26,29]. We tested the survival of cells at various time points following the initiation of curcumin treatment, including days 3, 6, 13, and 20. The viability of postmitotic cells was quantified through the outgrowth method [26,29] and normalized in relation to the baseline data obtained on day 3 (Figure 1A–D).

On day 6, both curcumin-treated and untreated postmitotic cells exhibited approximately 100% survival rates (Figure 1B). However, a notable contrast emerged on day 13. Postmitotic cells subjected to curcumin treatment, specifically at a concentration of 25 µM, displayed a striking enhancement in their survival rates, increasing to approximately 90%. In stark contrast, cells within the control group, devoid of curcumin treatment (receiving only DMSO), demonstrated a miserable survival rate, falling below the 10% threshold (Figure 1C). The divergence in survival dynamics persisted, substantiated by our observations on day 20, wherein curcumin-treated postmitotic cells maintained a robust survival rate of approximately 65% (Figure 1D).

These findings suggest that curcumin possesses the remarkable capacity to forestall cellular aging, and consequently, extend the PoMiCL of yeast. Our outcomes are consistent with an amount of prior research that has underscored the beneficial effects of curcumin in the context of cellular longevity [32,33,34]. Furthermore, our insights into curcumin’s influence on cellular lifespan complement analogous investigations conducted using different genetic backgrounds and distinct cell survival assays [35]. Collectively, these results compellingly advocate for the potential utility of curcumin as a modulator of cellular aging and lifespan extension across diverse eukaryotic species.

Intriguingly, our study revealed a complex dosage-dependent phenomenon. The maximal benefits of curcumin with regard to cellular lifespan extension were achieved at lower concentrations, whereas higher concentrations demonstrated diminishing efficacy (Figure 1C,D). This intriguing observation suggests a biphasic dose–response relationship, indicative of a hormetic effect exerted by curcumin on cellular lifespan.

### 2.2. Curcumin Increases Postmitotic Cellular Lifespan of Mitochondrial Dysfunctional Cells during Chronological Aging in Yeast

In our previous study, we demonstrated that inducing mitochondrial dysfunction through chemical or genetic means accelerates cellular aging and reduces the lifespan of postmitotic cells [36]. Given the pivotal role of mitochondrial function in age-related diseases and the potential influence of curcumin on cellular aging, we explored the impact of curcumin on the lifespan of postmitotic cells deficient in mitochondrial function during chronological aging. We assessed the effects of curcumin on the PoMiCL of mitochondrial gene-deletion mutants, associated with the loss of respiratory activity and energy synthesis. *PET100* (*YDR079W*) is a yeast gene critical for assembling and maintaining the mitochondrial respiratory chain complex IV (cytochrome c oxidase), which is essential for mitochondrial ATP production and overall cellular energy metabolism [37]. We conducted PoMiCL experiments on *pet100∆* mutants treated with various curcumin concentrations and compared them with wildtype cells. The survival of aging postmitotic cells was measured at specified time intervals relative to the outgrowth on day 3 (Figure 2A–D).

On day 5 and day 6, the survival of the *pet100∆* mutant (DMSO control) was below 10% (Figure 2B,C). Conversely, at the same chronological aging time points (day 5 and day 6), the survival rate of the wildtype (DMSO control) remained at approximately 100% (Figure 2B,C). These results align with our previous findings, illustrating that mitochondrial dysfunction leads to accelerated aging and a shortened PoMiCL [36]. Remarkably, curcumin treatment prevents accelerated aging and enhances the PoMiCL of the mitochondrial-dysfunction *pet100∆* mutant (Figure 2B,C). The survival rate of *pet100∆* cells treated with curcumin (50 µM) was around 90%, in contrast to less than 10% in untreated cells (Figure 2B). Interestingly, the survival rate increased with higher concentrations (100 µM and 200 µM) of curcumin, achieving nearly 100% viability on day 5 (Figure 2B). A similar concentration-dependent trend was observed for increased lifespan on day 6. The survival rate of *pet100∆* cells treated with the minimal effective curcumin concentration (100 µM) was approximately 60%, compared to less than 10% in untreated cells (Figure 2C). A higher curcumin concentration (200 µM) further boosted the survival of *pet100∆* cells to roughly 80% on day 6 (Figure 2C). These findings indicate that curcumin represents a promising anti-aging intervention that mitigates accelerated cellular aging and prolongs the lifespan of mitochondrial-dysfunction postmitotic cells.

Curcumin exhibits a biphasic dose–response pattern regarding the PoMiCL of the wildtype yeast strain, with the most potent anti-aging effects observed at a low concentration (25 μM), while its anti-aging activity diminishes at higher concentrations (50 μM, 100 μM, and 200 μM) (Figure 1C,D and Figure 2D). Interestingly, there was no discernible hormetic effect of curcumin on the PoMiCL of the mitochondrial-dysfunction *pet100∆* mutant (Figure 2B,C). These findings suggest that the hormetic impact of curcumin on PoMiCL is contingent upon mitochondrial functionality. To further validate this, we investigated the effect of curcumin on other mutant cells with mitochondrial dysfunction. *RMD9* (*YGL107C*) plays an important role in respiration and mitochondrial genome stability [38,39], controlling the expression of critical mitochondrial genes, including *COX2*, *CYTB*, and *ATP6*, which are integral components of the OXPHOS system. Conducting PoMiCL experiments on *rmd9∆*, along with *pet100∆* and wildtype strains, we observed a similar influence of curcumin on the PoMiCL of the *rmd9∆* mutant, similar to that observed in the *pet100∆* mutant (Figure 2B,C). Our findings indicate that curcumin in postmitotic cells afflicted with mitochondrial dysfunction (i) reverses cellular aging and prolongs lifespan, (ii) exhibits enhanced anti-aging activity with increasing concentrations, and (iii) lacks a hormetic effect on cellular lifespan.

By carefully scrutinizing all the data, we noticed that curcumin’s effective anti-aging concentrations (50 μM, 100 μM, and 200 μM) in mitochondrial mutants (Figure 2B,C) correspond to hormetic concentrations in wildtype cells (Figure 2D). Conversely, the curcumin concentration (25 μM) effective at promoting anti-aging in wildtype cells (Figure 2D) fails to enhance the survival of mitochondrial mutants (Figure 2B,C). This analysis indicates that curcumin’s anti-aging properties are influenced by both mitochondrial function and independent mechanisms. As a result, in the absence of anti-aging activity dependent on mitochondria, mitochondrial mutants experience a decreased capacity to extend cellular lifespan for an extended duration, in contrast to wildtype postmitotic cells, where both mitochondrial-dependent and -independent mechanisms come into play.

### 2.3. Curcumin Inhibits TORC1 and Prevents Accelerated Aging in Postmitotic Cells with Mitochondrial Dysfunction

Target of rapamycin complex 1 (TORC1) is a highly conserved master regulator of cellular metabolism [40,41,42,43]. It plays a positive role in anabolic processes while negatively impacting catabolic processes. TORC1 is a critical target for extending cellular lifespan, with its inhibition having been shown to increase healthspan and lifespan in a range of organisms, from yeast to animals [20,29]. TORC1 senses nutrients and is closely linked to the growth and proliferation of cells. Previously, we discovered that inhibiting TORC1 can prevent the accelerated cellular aging associated with mitochondrial dysfunction [36]. Interestingly, mutants with mitochondrial dysfunction were found to be resistant to rapamycin-induced cellular growth inhibition [36]. Furthermore, mutants such as *pet100∆* and *rmd9∆* also showed resistance to the growth-inhibiting effects of curcumin (Figure 3A). These results suggest that curcumin inhibits TORC1 activity, thereby preventing accelerated aging in cells with mitochondrial function deficiencies. To confirm this, we examined the effect of curcumin on TORC1 activity by monitoring the phosphorylation of the Sch9 substrate through western blot analysis [44,45]. Wildtype cells were treated with varying concentrations of curcumin, and Sch9 phosphorylation was assessed. We observed that curcumin efficiently inhibited TORC1 activity, similar to rapamycin treatment (Figure 3B; Appendix A). Additionally, the expression of the mitochondrial *SDH1* gene was similarly affected by both curcumin and rapamycin treatments (Figure 3C). These results collectively confirm that curcumin inhibits TORC1 activity.

AMP-activated protein kinase (AMPK) is known to negatively regulate the TORC1 activity pathway [46,47,48]. Notably, AMPK promotes catabolic processes while inhibiting anabolic pathways, contrasting with the TORC1 signaling pathway. In the yeast *Saccharomyces cerevisiae*, SNF1 protein kinase serves as the ortholog of the human AMPK complex [49,50,51]. We were curious whether curcumin’s inhibition of TORC1 was dependent on the AMPK pathway. To investigate this, we assessed the effect of curcumin on TORC1 activity in *snf1∆*-deletion cells. We found that the inhibition of Sch9 phosphorylation was comparable in the *snf1∆* mutant compared to the wildtype (Figure 3D; Appendix A). This result indicates that curcumin’s inhibition of TORC1 activity is independent of the AMPK pathway. Altogether, our findings reveal that curcumin inhibits TORC1 activity and rescues mitochondrial dysfunction.

### 2.4. Curcumin Increases Mitochondrial Function and Oxidative Stress

We further investigated the effect of curcumin on mitochondrial function by measuring cellular energy levels. Wildtype cells were supplemented with varying concentrations of curcumin and total cellular ATP was measured. Curcumin significantly increases ATP levels in cells (Figure 4A). Next, we tested the effect of curcumin on the expression of mitochondrial electron transport chain (ETC) genes (*SDH2*, *COX6*, *COX7*, and *ATP6*) [52]. We found that curcumin increases the expression of the tested ETC genes (Figure 4B). Together these results reveal that curcumin enhances the mitochondrial function.

Mitochondrial function is closely linked to the generation of reactive oxygen species (ROS) in cells [53,54,55,56]. Superoxide dismutase 2 (SOD2) is a highly conserved mitochondrial antioxidant enzyme that protects against oxidative damage induced by reactive oxygen species (ROS) [53,54,55,56]. To assess the impact of curcumin on oxidative stress, we analyzed the expression of the *SOD2* gene. Curcumin treatment resulted in an increase in the expression of the *SOD2* gene (Figure 4C). This finding indicates that curcumin induces oxidative stress. Additionally, we investigated the effect of curcumin on the expression of cytoplasmic superoxide dismutase 1 (*SOD1*). We observed that *SOD1* expression was comparable in both curcumin-treated and non-treated cells (Appendix A). Collectively, these results suggest that curcumin specifically induces mitochondrial oxidative stress.

Since curcumin increases mitochondrial function and oxidative stress, we aimed to verify its phenotypic effect on cellular growth. In our yeast strain background, we initially tested whether *SOD2* is involved in oxidative stress. We exposed wildtype and *sod2∆*-deletion cells to the oxidative stress-inducing chemical hydrogen peroxide (H_2_O_2_) and assessed their growth. Our results indicated that the *sod2∆* mutant was more sensitive to oxidative stress compared to the wildtype (Figure 4D). It is known that oxidative stress is associated with an increase in cellular aging [53,54,55,56]. We also observed that the PoMiCL of the *sod2∆* mutant was shorter than that of the wildtype (Figure 4E). These findings established that *sod2∆* mutant cells are more vulnerable to oxidative stress in our experimental yeast genetic background.

To further investigate whether curcumin induces oxidative stress, we examined the growth of *sod2∆* mutant cells compared to wildtype cells when exposed to curcumin. Our results showed that the growth of the curcumin-treated *sod2∆* mutant was lower than that of wildtype cells (Figure 4F). This outcome suggests that curcumin indeed induces oxidative stress, making *sod2∆* cells more sensitive to its effects. These results align with previous findings that have demonstrated increased ROS production following curcumin treatment in cells [35]. Therefore, our results indicate that curcumin enhances ATP levels but also induces oxidative stress in cells. While the increase in energy may be associated with an extended cellular lifespan, the associated oxidative stress may counteract the beneficial effects of curcumin, eliciting a hormetic response in the cells.

## 3. Discussion

This study investigates the potential of curcumin, a polyphenolic compound derived from turmeric [22,23,24,25], in extending the lifespan of postmitotic cells during chronological aging, especially in the context of mitochondrial dysfunction. Our results demonstrate that curcumin has a significant impact on extending the lifespan of postmitotic cells during chronological aging. These findings align with previous studies that have highlighted the beneficial effects of curcumin on healthspan and cellular longevity [32,33,34,57,58]. Notably, curcumin’s anti-aging effects exhibit a biphasic dose–response pattern, with lower concentrations showing greater efficacy.

Mitochondrial dysfunction is a key factor associated with age-related diseases, including cardiovascular diseases, neurodegenerative disorders, sarcopenia, and age-related macular degeneration [5,6,7,8]. We revealed that curcumin not only extends the lifespan of the wildtype but also rescues the shortened lifespan of mitochondrial-defective postmitotic cells during chronological aging in yeast. This finding suggests that curcumin may have therapeutic potential in mitigating the effects of mitochondrial dysfunction, a common hallmark of aging [59,60].

Our investigation into the mechanisms underlying curcumin’s anti-aging properties revealed that curcumin inhibits TORC1 activity. TORC1 is a central regulator of cellular metabolism and aging [40,41,42,43], and its inhibition has been linked to increased lifespan across various organisms [20,29]. Curcumin’s ability to inhibit TORC1 activity is crucial for its anti-aging effects, especially in cells with mitochondrial dysfunction. Curcumin enhances mitochondrial function by increasing ATP levels. This improvement in energy production is associated with an extension of cellular lifespan. However, it is important to note that curcumin also induces oxidative stress, potentially as a byproduct of increased ATP synthesis. The interplay between enhanced energy production and oxidative stress may contribute to the hormetic response observed with curcumin treatment.

The results of this study have several important implications for the field of aging research. First, curcumin derived from natural sources such as turmeric-rich diets may have the potential to promote healthy aging and extend lifespan. Second, curcumin’s ability to rescue postmitotic cells with mitochondrial dysfunction highlights its relevance for age-related diseases associated with impaired mitochondrial function. Third, the hormetic response observed with curcumin treatment underscores the importance of dosage and further emphasizes the need for careful consideration when designing interventions. In summary, our study provides valuable insights into the intricate relationship between curcumin, postmitotic cellular aging, TORC1 activity, and mitochondrial function.

## 4. Methods

### 4.1. Yeast Strains and Culture Conditions

We utilized the prototrophic CEN.PK113-7D strain [61] of *Saccharomyces cerevisiae* for our experiments. Gene deletions and protein tagging were accomplished through the application of conventional PCR-based techniques [62,63]. To initiate yeast experiments, strains were resuscitated from frozen glycerol stocks and incubated on YPD agar medium, which comprised 1% Bacto yeast extract, 2% Bacto peptone, 2% glucose, and 2.5% Bacto agar. Incubation was conducted at a temperature of 30 °C for a duration of 2–3 days.

### 4.2. Chemical Treatments of Cell Cultures

For the introduction of chemical treatments, we prepared stock solutions of curcumin and rapamycin, employing dimethyl sulfoxide (DMSO) as the solvent. The final concentration of DMSO in the yeast experiments was consistently maintained below 1%.

### 4.3. Analysis of Postmitotic Cellular Lifespan during Chronological Aging in Yeast

The lifespan of yeast postmitotic cells was assessed by determining the survival in chronological aging, as previously described [29,36]. Yeast cultures were grown in synthetic defined (SD) medium (6.7 g/L yeast nitrogen base with ammonium sulfate, without amino acids, and with 2% glucose) overnight at 30 °C with 220 rpm shaking. The cultures were then diluted to an initial optical density at 600 nm (OD600) of approximately 0.2 in fresh SD medium to initiate the postmitotic cellular lifespan (PoMiCL) experiment. In short, cell cultures were cultivated in 96-well plates and allowed to enter the postmitotic phase at 30 °C. At various chronological time points, the survival of aging postmitotic cells was determined by measuring the outgrowth (OD600nm) in YPD medium incubated for 24 h at 30 °C, using a microplate reader.

### 4.4. Measurement of ATP Level

For the analysis of ATP (adenosine triphosphate), yeast cells were initially treated with a 5% trichloroacetic acid (TCA) solution and chilled on ice for at least 5 min to preserve cellular contents. Subsequently, the cells underwent washing to remove the extracellular residual before being suspended in a 10% TCA solution to aid in cellular disruption for ATP extraction. Mechanical disruption, facilitated by glass beads in a bead beater, was employed to release ATP into the solution. ATP quantification was performed using the PhosphoWorks™ Luminometric ATP Assay Kit (AAT Bioquest), based on a chemical reaction generating light proportional to ATP concentration [45]. To ensure accurate comparisons, ATP levels were normalized against the protein content, determined using the Bio-Rad protein assay kit.

### 4.5. RNA Extraction and qRT-PCR

Total RNA isolation from yeast cells was performed following a previous study method [36,44]. RNA was isolated using the Qiagen RNeasy mini kit by mechanically disrupting the cells according to the manufacturer’s guidelines. Subsequently, the concentration and integrity of the RNA were assessed using the ND-1000 UV–visible light spectrophotometer from Nanodrop Technologies and the Bioanalyzer 2100 with the RNA 6000 Nano Lab Chip kit by Agilent. Quantitative reverse transcription polymerase chain reaction qRT-PCR experiments were performed as described previously using the QuantiTect Reverse Transcription Kit (Qiagen) and SYBR Fast Universal qPCR Kit (Kapa Biosystems) [36,44]. The abundance of each gene was determined relative to the house-keeping transcript *ACT1*.

### 4.6. Oxidative Stress Growth Sensitivity Assay

The assay was conducted to investigate the impact of oxidative stress-inducing agents on cell growth using 96-well plates. Yeast cells, with an initial OD600 of approximately 0.2 in SD medium, were transferred into the 96-well plate containing serially double-diluted concentrations of hydrogen peroxide (H_2_O_2_) or curcumin. The cells were then incubated at 30 °C, and their growth was monitored by measuring OD600 using a microplate reader.

### 4.7. Analysis of TORC1 Activity

TORC1 activity experiments were conducted as previously described [44,45]. Protein samples were separated using SDS-PAGE and transferred onto nitrocellulose membranes for western blotting. The membranes were blocked with 5% milk in TBS/0.1% Tween 20 solution. Phosphorylation of Sch9 was assessed using an anti-HA 3F10 antibody (dilution: 1:2000; Roche Life Science, USA), followed by incubation with a goat anti-rat HRP-conjugated antibody (dilution: 1:5000; Santa Cruz Biotechnology). Blots were developed using ECL Prime western blotting detection reagent (Amersham Pharmacia Biotech, USA) and quantified using ImageJ with the iBright CL1500 Imaging System (Thermo-Scientific).

### 4.8. Data Quantification and Statistical Interpretation

Statistical analyses of the results, including calculations of mean values, standard deviations, significance testing, and graphing, were conducted using GraphPad Prism v.10 software. Data comparisons were statistically evaluated using Student’s *t*-tests, ordinary one-way ANOVA, and two-way ANOVA, followed by post hoc multiple comparison tests. In all graph plots, significance levels are represented as * *p* < 0.05, ** *p* < 0.01, *** *p* < 0.001, and **** *p* < 0.0001. Data that did not reach significance are denoted as ‘ns’ for non-significant.

## Figures and Tables

**Figure 1 cells-13-01470-f001:**
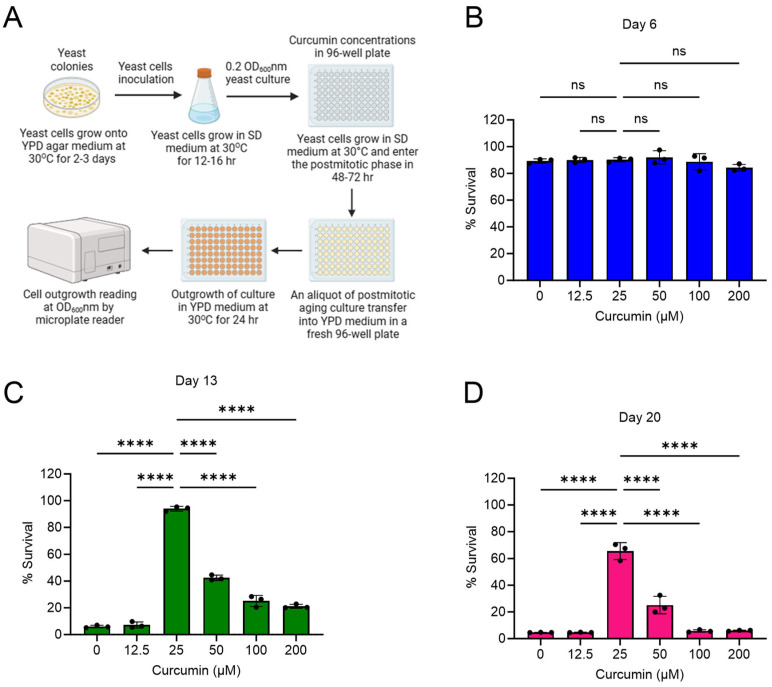
Curcumin prolongs the lifespan of postmitotic cells during chronological aging in yeast. (**A**) Schematic of the study workflow. The effect of curcumin treatment on the postmitotic cellular lifespan (PoMiCL) of yeast *Saccharomyces cerevisiae* genetic background CEN.PK113-7D wildtype was evaluated in an SD medium using a 96-well plate. Cells were incubated with indicated concentrations of curcumin in the 96-well plate. The survival of chronological aging postmitotic cells was measured on (**B**) day 6, (**C**) day 13, and (**D**) day 20, relative to the outgrowth of day 3. The data are presented as means ± SD (n = 3). Statistical significance was determined as follows: **** *p* < 0.0001 and ns (non-significant), based on a two-way ANOVA followed by Dunnett’s multiple comparisons test.

**Figure 2 cells-13-01470-f002:**
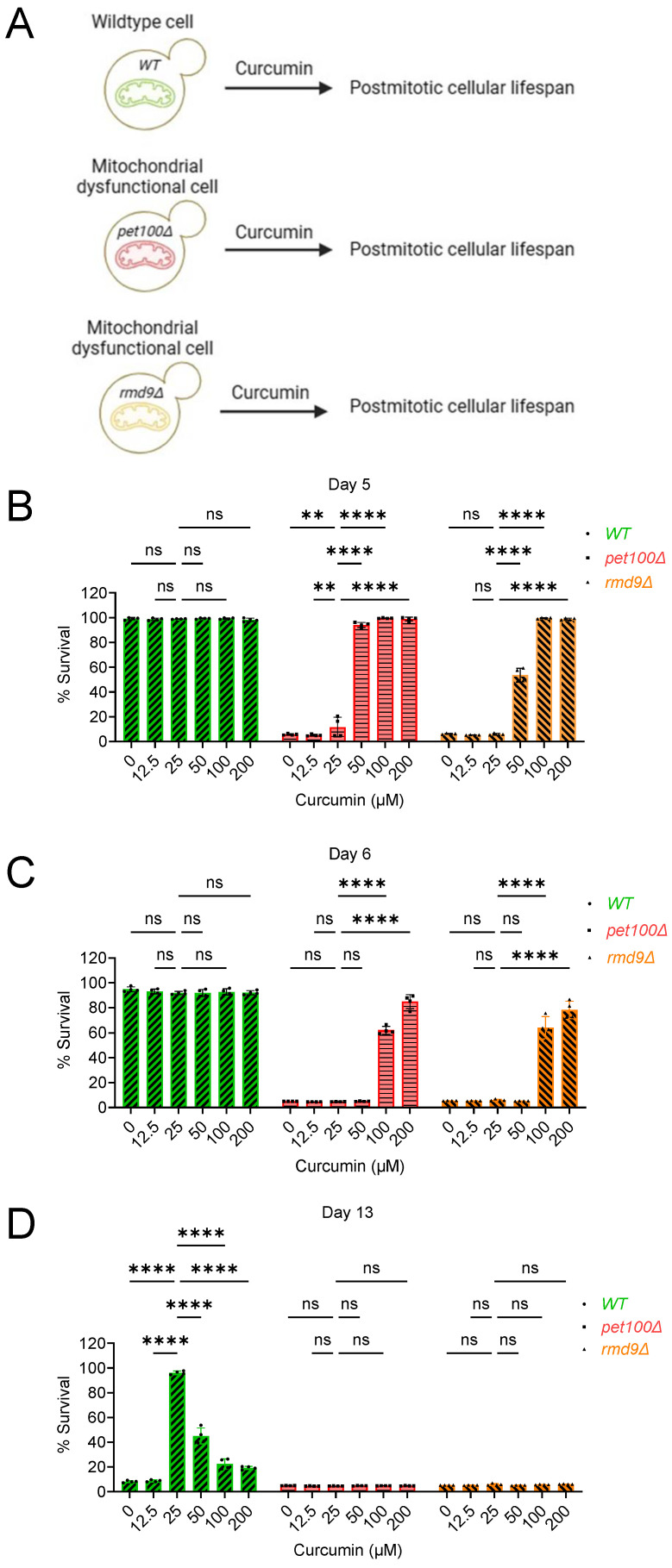
Curcumin prolongs the postmitotic cellular lifespan of mitochondrial dysfunctional cells during chronological aging in yeast. (**A**) Schematic of the study workflow. The effect of curcumin treatment on the postmitotic cellular lifespan (PoMiCL) of yeast *Saccharomyces cerevisiae* genetic background CEN.PK113-7D wildtype and mitochondrial-dysfunction mutants (*pet100∆* and *rmd9∆*) was evaluated in an SD medium using a 96-well plate. Cells were incubated with indicated concentrations of curcumin in the 96-well plate. The survival of chronological aging postmitotic cells was measured on (**B**) day 5, (**C**) day 6, and (**D**) day 13, relative to the outgrowth of day 3. The data are presented for replicates (n = 4). Statistical significance was determined as follows: ** *p* < 0.01, **** *p* < 0.0001 and ns (non-significant), based on a two-way ANOVA followed by Dunnett’s multiple comparisons test.

**Figure 3 cells-13-01470-f003:**
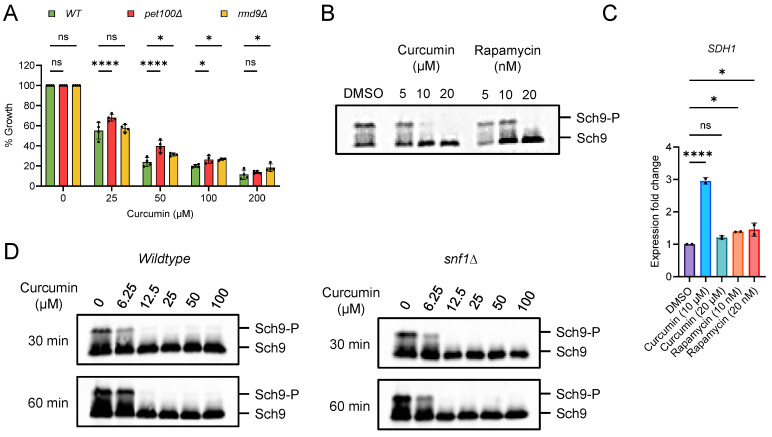
Curcumin inhibits the TORC1 signaling pathway. (**A**) The growth assay was conducted using yeast *Saccharomyces cerevisiae* genetic background CEN.PK113-7D wildtype cells and mitochondrial-dysfunction mutants (*pet100∆* and *rmd9∆*). They were treated with varying concentrations of curcumin for 72 h. Growth rates were normalized to the untreated control. The data are presented as means ± SD (n = 4). Statistical significance was determined as follows: * *p* < 0.05, **** *p* < 0.0001, and ns (non-significant) based on a two-way ANOVA followed by Dunnett’s multiple comparisons test. (**B**,**C**) Yeast exponential cultures of wildtype Sch9-6xHA-Tag cells were exposed to varying concentrations of curcumin and rapamycin for one hour. Subsequently, aliquots from these cultures were collected to extract proteins and RNA. (**B**) TORC1 activity was assessed by monitoring the phosphorylation of the Sch9 substrate through western blotting. (**C**) Expression analysis of *SDH1* gene was analyzed by qRT-PCR of yeast cells treated with DMSO, curcumin, and rapamycin. Gene expression of curcumin- and rapamycin-treated samples were compared with DMSO control. Data are represented as means ± SD (n = 2). (**B**) * *p* < 0.05, **** *p* < 0.0001, and ns (non-significant) based on an ordinary one-way ANOVA followed by Dunnett’s multiple comparisons test. (**D**) Wildtype and *snf1∆* exponential cultures of Sch9-6xHA-Tag were treated with different concentrations of curcumin for the indicated times. Aliquots of the cultures were collected to prepare protein extracts, and TORC1 activity was assessed by monitoring the phosphorylation of the substrate, Sch9, through western blotting.

**Figure 4 cells-13-01470-f004:**
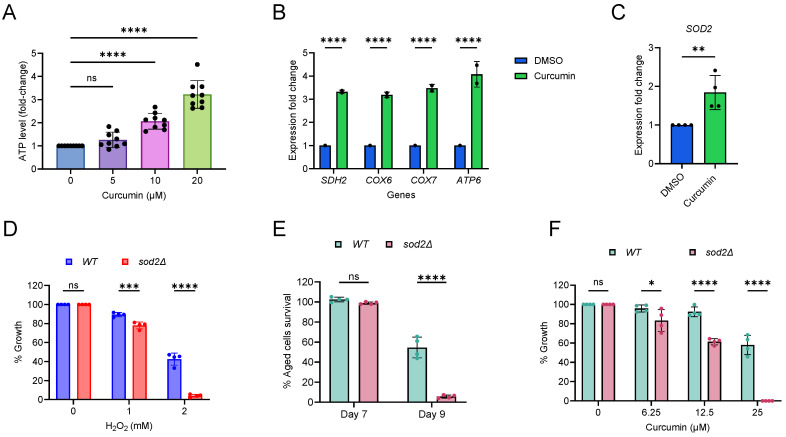
Curcumin enhances ATP levels and oxidative stress. (**A**) ATP analysis of yeast *Saccharomyces cerevisiae* genetic background CEN.PK113-7D wildtype cells treated with indicated concentrations of curcumin for one hour. The data are presented as means ± SD (n = 9). **** *p* < 0.0001, as determined by an ordinary one-way ANOVA followed by Dunnett’s multiple comparisons test. (**B** and **C**) Expression of (**B**) mitochondrial ETC genes and (**C**) *SOD2* gene was analyzed by qRT-PCR of yeast cells treated with DMSO and curcumin (10 μM). Gene expression of curcumin-treated samples were compared with DMSO control. (**B**) Data are represented as means ± SD (n = 2) and **** *p* < 0.0001 based on a two-way ANOVA followed by Šídák’s multiple comparisons test. (**C**) Data are represented as means ± SD (n = 4) and ** *p* < 0.01 based on two-sided Student’s *t*-tests. (**D**,**F**) Growth assay of yeast wildtype cells exposed to (**B**) hydrogen peroxide (H_2_O_2_) and (**C**) curcumin at the indicated concentrations for 72 h. Growth was normalized to the untreated control. The data are presented as means ± SD (n = 4). Statistical significance was determined as follows: * *p* < 0.05, *** *p* < 0.001, **** *p* < 0.0001, and ns (non-significant) based on a two-way ANOVA followed by Šídák’s multiple comparisons test. (**E**) Lifespan of yeast wildtype and *sod2∆*-deletion postmitotic cells was evaluated in an SD medium using a 96-well plate. The survival of aging postmitotic cells was measured on day 7 and day 9, relative to the outgrowth of day 3. The data are presented as means ± SD (n = 4). Statistical significance was determined as follows: **** *p* < 0.0001 and ns (non-significant), based on a two-way ANOVA followed by Šídák’s multiple comparisons test.

## Data Availability

Further information and requests for data, resources and reagents should be directed to and will be fulfilled by the Lead Contact, Mohammad Alfatah (alfatahm@bii.a-star.edu.sg or alfatahm@nus.edu.sg).

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
