# Peer review of "Curcumin Inhibits TORC1 and Prolongs the Lifespan of Cells with Mitochondrial Dysfunction"

_cells, 2024, doi:10.3390/cells13171470_

Round 1

Reviewer 1 Report

Comments and Suggestions for Authors

The article presented by Naaz and collaborators studies the effect of curcumin on prolonging lifespan of cell in conditions of mitochondrial dysfunction. Using a model with Saccharomyces cerevisiae, they were able to demonstrate that the use of curcumin prolongs the postmitotic life of this yeast, even when they carry mitochondrial damage. I find this work interesting, however generalizing this effect to all types of eukaryotic cells seems incorrect to me. The results in HEK 293 cells are insufficient to conclude that both phenomena are homologous.

On the other hand, curcumin is a compound with antioxidant capacity widely reported in the literature, I do not see the novelty of this work. Curcumin is a very good antioxidant compound, but its solubility is very low in aqueous solutions, this is a significant challenge. On the other hand, it is widely reported that the inhibition of TORC1 favors the autophagy process, allowing cells that enter a state of senescence to survive using their own metabolic substrates.

Figure 3 and 4 are inverted.

Author Response

Dear Reviewer,

Thank you for your insightful comments on our manuscript. We have carefully considered your feedback and made the following revisions:

In response to your concerns about generalizing the effects observed in HEK 293 cells, we have decided to remove these results from our manuscript. The focus of our study will now be specific to the yeast aging model, Saccharomyces cerevisiae, to provide a more targeted and accurate analysis of curcumin's effects on postmitotic cells with mitochondrial deficiencies.

We agree with your observation that curcumin's role in TORC1 inhibition and its downstream processes is widely reported. However, our study is the first to demonstrate that curcumin extends the lifespan of postmitotic cells specifically under conditions of mitochondrial dysfunction. Additionally, we discovered that the hormetic effect of curcumin, which is observed in wild-type cells, is absent in mitochondrial-deficient cells. This finding reveals that the hormetic phenotype of curcumin is dependent on mitochondrial function in these cells.

We believe these revisions will strengthen our manuscript by providing a clearer focus and highlighting the novel contributions of our work.

We have corrected the inversion of Figures 3 and 4 in the revised manuscript. Thank you for bringing this to our attention.

Thank you once again for your valuable feedback.

Reviewer 2 Report

Comments and Suggestions for Authors

After a detailed evaluation, analyzing the scientific relevance of these findings, this manuscript should be accepted after minor revisions. The text is fine in grammar and language spelling, but there are some points that must be solved:

1.    Abstract: incomplete; a short sentences must be included in introduction, justifying this study. The results must be also shortly described.

2.    Introduction section: must be improved; poor contextualization; the authors must better describe the state of art of curcumin, especially.

3.    Methods: Is not necessary to synchronize the cells before post mitotic analysis? In relation TORC analysis, do the authors think SDS-PAGE assess activity? I really don´t think so…

4.    IMPORTANT: figures 3 and 4 were wrong. Figure 3 is figure 4, and figure 4 is figure 3… Please correct it.

5.    Results section: the authors included many information that is not results here. Too much discussion in the wrong section. Once there is a specific discussion section, all discussion must be there, cleaning the results. Please correct it.

Author Response

Dear Reviewer,

Thank you for your thorough evaluation and valuable suggestions for our manuscript. We appreciate your positive feedback and have made the necessary revisions to address your concerns:

We have revised the abstract to include a brief introduction justifying the study, as well as a concise summary of the key results. This should provide a clearer overview of our study's purpose and findings.

The introduction has been expanded to provide a better contextualization of curcumin research. We have included a more detailed of the current state of the art regarding curcumin, highlighting its relevance and the unique aspects our study brings to the field.

The lifespan of yeast postmitotic cells was assessed by determining their survival during chronological aging, as previously described in references 29 and 36. This is a well-established method. Mitotic cells naturally synchronized into postmitotic cells after 24-48 hours before we began determining their chronological lifespan.

For TORC1 analysis, its activity was conducted as described in references 44 and 45. This is also a well-established method, where we used SDS-PAGE to separate the proteins, followed by transfer onto nitrocellulose membranes for Western blotting

We have corrected the inversion of Figures 3 and 4 in the revised manuscript. Thank you for pointing this out.

We have cleaned up the Results section by removing any discussion content that was previously included.

We believe these revisions improve the clarity and quality of our manuscript. Thank you once again for your insightful feedback.

Round 2

Reviewer 1 Report

Comments and Suggestions for Authors

Dear authors,

Thank you for considering my comments. I think the new version is more appropriate, removing the experiments on HEK cells allows a better understanding of the objective of the work. The findings regarding mitochondrial dysfunction are interesting, personally, I would continue with this line of research.